# Structure-based 3D-Pharmacophore modeling to discover novel interleukin 6 inhibitors: An in silico screening, molecular dynamics simulations and binding free energy calculations

Que-Huong Tran[1,2], Quoc-Thai Nguyen[1]*, Nguyen-Quynh-Huong Vo[1], Tan Thanh Mai[1], Thi-Thuy-Nga Tran[1,2], Thanh-Dao Tran[1], Minh-Tri Le[1,3], Dieu-Thuong Thi Trinh[4]*, Khac-Minh Thai[1]*

1 Faculty of Pharmacy, University of Medicine and Pharmacy at Ho Chi Minh City, Ho Chi Minh City, Vietnam, 2 Department of Pharmaceutical Chemistry Da Nang University of Medical Technology and Pharmacy, Da Nang, Vietnam, 3 School of Medicine, Vietnam National University Ho Chi Minh City, Linh Trung Ward., Thu Duc Dist., Ho Chi Minh City, Vietnam, 4 Faculty of Traditional Medicine, University of Medicine and Pharmacy at Ho Chi Minh City, Ho Chi Minh City, Vietnam

* nqthai@ump.edu.vn (QTN); thuong.ttd@ump.edu.vn (DTTT); thaikhacminh@ump.edu.vn, thaikhacminh@gmail.com (KMT)

**Data Availability Statement:** All relevant data are within the manuscript and its Supporting Information files.

## Abstract

Interleukin 6 (IL-6) is a cytokine with various biological functions in immune regulation, hematopoiesis, and inflammation. Elevated IL-6 levels have been identified in several severe disorders such as sepsis, acute respiratory distress syndrome (ARDS), and most recently, COVID-19. The biological activity of IL-6 relies on interactions with its specific receptor, IL-6Rα, including the membrane-bound IL-6 receptor (mIL-6R) and the soluble IL-6 receptor (sIL-6R). Thus, inhibition of the interaction between these two proteins would be a potential treatment for IL-6 related diseases. To date, no orally available small-molecule drug has been approved. This study focuses on finding potential small molecules that can inhibit protein-protein interactions between IL-6 and its receptor IL-6Rα using its crystal structure (PDB ID: 5FUC). First, two pharmacophore models were constructed based on the interactions between key residues of IL-6 (Phe74, Phe78, Leu178, Arg179, Arg182) and IL-6Rα (Phe229, Tyr230, Glu277, Glu278, Phe279). A database of approximately 22 million compounds was screened using 3D-pharmacophore models, molecular docking models, and ADMET properties. By analyzing the interactive capability of successfully docked compounds with important amino acids, 12 potential ligands were selected for further analysis via molecular dynamics simulations. Based on the stability of the complexes, the high interactions rate of each ligand with the key residues of IL-6/IL-6Rα, and the low binding free energy calculation, two compounds ZINC83804241 and ZINC02997430, were identified as the most potential IL-6 inhibitor candidates. These results will pave the way for the design and optimization of more specific compounds to combat cytokine storm in severe coronavirus patients.

**Funding:** This work was supported by the VietNam National Foundation for Science and Technology Development (NAFOSTED) under the Grant Number 108.05-2018.15 (to Quoc-Thai Nguyen) [https://nafosted.gov.vn/]. Tan Thanh Mai was funded by Vingroup Joint Stock Company and supported by the Domestic Master/ PhD Scholarship Programme of Vingroup Innovation Foundation (VINIF), Vingroup Big Data Insti-tute (VINBIGDATA), code VINIF.2020.TS.128 [https://vinbigdata.org/quy-vinif/]. The funders had no role in study design, data collection and analysis, decision to publish, or preparation of the manuscript.

**Competing interests:** The authors have declared that no competing interests exist.

## Introduction

Interleukin 6 (IL-6) is a member of the IL-6 family of cytokines that includes IL-11, IL-27, IL-31, leukocyte-inhibitory factor, oncostatin M, cardiotrophin-1, and ciliary neurotrophic factor [1, 2]. The signaling pathway of IL-6 is initiated by the formation of a complex between IL-6 and its specific receptor (IL-6Rα). Subsequently, the complex bind to a cell-surface glycopro-tein130 (gp130), which is shared among the IL-6 family of cytokines. The aberrant production of IL-6 and its receptor has been implicated in the pathogenesis of multiple myeloma, post-menopausal osteoporosis, rheumatoid arthritis, and autoimmune diseases [3]. After forming a complex between IL-6 and the membrane or soluble form of IL-6Rα (mIL-6Rα or sIL-6Rα), several modes of gp130 activation are initiated [4, 5]. Classic signaling mediated by IL-6 and mIL-6Rα plays a significant role in the acute-phase immunological response and promotes anti-inflammatory activities, whereas binding of IL-6 to sIL-6Rα induces pro-inflammatory trans-signaling [4, 6].

In the past two years, the COVID pandemic outbreak has killed approximately 6 million patients and there were over 400 million cases of infections [7]. IL-6 and IL-6Rα were also investigated for their association with this infectious disease. It has been suggested that one possible mechanism underlying rapid disease progression is a cytokine storm. Reports on the immunological profile of critically ill patients with COVID-19 indicate that elevated levels of interleukin-6 are associated with respiratory failure, shock, and multiorgan dysfunction [8, 9]. Therefore, modulating the levels of IL-6 or its effects is a highly critical point and therapeutic target for COVID-19 patients.

IL-6 activates cells by binding to IL-6 receptor (IL-6Rα) and gp130. Ternary complex forms a hexamer consisting of two molecules each of IL-6, IL6Rα, and gp130 chains, sequential and cooperative assembly. IL-6 must first form a complex with a non-signaling receptor, IL-6Rα, through surface regions on both proteins defined as site I. The site II (site IIa and site IIb) is formed by the binary complex of IL-6/ IL-6Rα that would interact with the D2 and D3 domains of gp130, fortuitously contributing to the stabilization of the IL-6/IL-6Rα/gp130 sig-naling complex. The subsequent interaction of site III is formed by the immunoglobulin-like D1 domain of gp130 with IL-6 (site IIIa) and the D2 domain of IL-6Rα (site IIIb) [10]. These three distinct interaction surfaces allow the assembly of a stable, ternary signaling complex composed of two molecules of each component.

Several monoclonal antibodies (mAbs) blocking IL-6 or IL-6Rα are in clinical development, many of which block the IL-6 classic signaling and the IL-6 trans-signaling pathways. Particu-larly, tocilizumab–a humanized anti-IL-6R that inhibits both sIL-6Rα and mIL6Rα–was approved by the FDA for treating rheumatoid arthritis, idiopathic multicentric Castleman's disease (iMCD), and cytokine release syndrome [11]. Siltuximab, a chimeric anti-IL-6, has recently been licensed for the treatment of iMCD [12]. Other mAbs target IL-6Rα (sarilumab, satralizumab, vobarilizumab), IL-6 (olokizumab, sirukumab, clazakizumab) [12, 13]. However, the mAb has several drawbacks, such as high cost, invasive administration, and a high rate of immunogenicity. Small molecule therapy has several advantages over biological therapeutic agents, including easier oral administration, superior tissue penetration, modifiable pharma-cokinetics, and lower production costs. Most studies mainly focused on the design of targets that directly inhibit the gp130 D1 domain and block the IL-6/gp130/STAT3 signaling pathway such as madindoline A, SC144, bazedoxifene, raloxifene, and LTM-28 [14, 15].

To date, no orally available small molecule drug that inhibits the IL-6 or IL-6Rα is approved for the treatment of related diseases. The present study aims to discover small synthetic mole-cules as drug candidates for IL-6-mediated disorders by blocking the binding site I of the IL-6/ IL-6Rα complex. This would be highly valuable, especially for severe COVID-19 patients. In

this study, the pharmacophore models created based on key residues of IL-6 and IL-6Rα at their active site were first used to screen for lead compounds [16], after which ligands were docked in the binding site of IL-6/IL-6Rα to examine protein-ligand interactions. Finally, from 22 million substances from ZINC12, the best-docked compounds were selected for molecular dynamics simulations.

## Materials and methods

In silico models including 3D-pharmacophores and molecular docking were built for virtually screening a compound library from ZINC12 databases. The absorption, distribution, metabolism, elimination, and toxicity (ADMET) properties of compounds were predicted. Molecular dynamics (MD) simulation was applied to evaluate the stability of the docked complex and estimate the dynamic behavior of the protein-ligand complex,. Finally, the molecular mechanics Generalized Born surface area (MM-GBSA) approach was used to calculate the binding energy of the ligands with IL-6/IL-6Rα to assess the potential inhibitors. The procedure is outlined in Fig 1.

### 3D-Pharmacophore generation

The 3D-pharmacophore models based on the protein-protein interaction (PPI pharmacophore) were only established when a 3D structure of the PPI complex was available [7, 16]. Therefore, in this work, we used a manual PPI pharmacophore defined from the X-ray structure of IL-6/IL-Rα complex with the epitope antibody VHH6 at 2.7 Å resolution (PDB ID: 5FUC) [17]. The IL-6 interacts with its receptor IL-6Rα as a well-known example of a PPI. Still, the IL-6/IL-6Rα binary complex has a low binding affinity, while VHH6 has the ability to stabilize and modulate the interaction of these two proteins. By blocking IL-6 and IL-6Rα together, this protein complex structure is particularly suitable for drug discovery purposes. It could be an instrumental for screening more specific inhibitors of the IL-6/IL-6Rα complex, leading to selective inhibition of the IL-6 trans-signaling pathway in pathologies [18].

The IL-6 and IL-6Rα interacts with each other through only one binding site (site I). Most studies have shown that site I is indispensable for the recruitment of two gp130 signaling receptors at sites II and III. At binding site I, the IL-6/IL-6Rα interface was formed by hydrophobic regions surrounded by several hydrophilic amino acid clusters from both proteins. The original interactions between Phe74, Phe78, Leu178, Arg179 and Arg182 of IL-6 and Phe229, Tyr230, Glu277, Glu278 and Phe279 of IL-6Rα (Table 1) were used to design two pharmacophore models [19]. Residues that play a significant role (hot-spot residues) in this protein-protein interaction (PPI) have been identified by mutation studies [20]. This step was performed by the Pharmacophore Query Editor tool in the Molecular Operating Environment (MOE) 2015.10 software [21].

### Online virtual screening through pharmacophore models in ZINCPHARMER

A total of 21,777,093 compounds from ZINC12 were used for pharmacophore-based virtual screening using the ZINCPharmer webserver provided by ZINC's developer [22]. The database's freely accessible website interface presents purchasable compounds in 3D formats that are ready to dock. Nevertheless, the program has a simpler interface, attribute points and constraints than the MOE 2015.10. Therefore, the 3D-pharmacophore model, after being uploaded, needs to be adjusted by creating binding combinations manually. To easily predict and select the drug-like compounds, the obtained hits then were filtered based on the Lipinski's "Rule of Five". Under this law, drug-like properties must have a log P-value < 5, a

**ZINC12 Database**
(21,777,093 compounds)

**Pharmacophore models**
(MOE 2015.10, ZINCPharmer)

**ADMET Prediction**
(ADMET Predictor 10.0)

**Molecular docking**
(FlexX in LeadIT 2.1.8)

**MD simulation**
(Gromacs 2020.2, VMD)

**Binding energy**
(gmx_MMPBSA)

**Fig 1. Study flowchart.**

molecular weight < 500 Da, H-binding acceptors (HBA) ≤ 10 and H-binding donors (HBD) ≤ 5 [23].

The ADMET Predictor 10.0 software [24] was used to evaluate the compounds' absorption, distribution, metabolism, elimination, and toxicity properties compared to reference

**Table 1. Interaction between hot-spot residues of IL-6 and the corresponding residues of its receptor at the binding site I.**

| IL-6Rα residues | Type of interaction | IL-6 residues |
|---|---|---|
| Phe229 | Aromatic ring | Phe78 |
| Tyr230 | Aromatic ring | Phe74 |
| Phe279 | Hydrophobic | Leu178 |
| Glu277, Glu278 | (-) Salt bridge (+) | Arg179, Arg182 |

thresholds for risk values in preclinical and clinical predictions. In addition, adverse effects on genital organs, teratogenicity and developmental were also evaluated through the parameter Reproductive Toxicity. Reproductive toxicity is an important regulatory endpoint in health hazard assessment [25].

## Molecular docking

The FlexX tool of BioSolveIT LeadIT 2.1.8 software [26] was used to build molecular docking mode. First, the crystal structure of IL-6/IL-6Rα (PDB ID: 5FUC) was prepared by using the QuickPrep tool in the MOE software, including the steps of "add hydrogens", "protonate" (charge amino acids), "tether and minimize", "delete unbound waters", and "refine". Before being imported into the LeadIT software. The ligand databases that satisfied the 3D-pharmacophore models and ADMET predictions were minimized the energy by using MOE 2015.10 software with Amber10: EHT force field. The energy gradient was set to RMS of 0.0001 kcal/mol/A$^2$ to obtain ready-to-dock structures.

The docking functionality applies the flexible docking methodology to search for the ligand conformations and manipulate the experimental scoring to score and rank the docking poses based on an incremental construction algorithm [27]. At binding site I of IL-6/IL-6Rα, key residues within a radius of 10 Å were selected and marked. In this process, the ligands were split into fragments, and an initial fragment was placed into multiple places in the binding pocket and scored using a pre-scoring scheme. The maximum number of solutions per interation was set to 1000, the maximum number of solutions per fragmentation was 200, and the number of poses to keep was Top10 [28]. Docking scores of successfully docked ligands were evaluated; their interactions to hot-spot residues were recorded. The Protein-Ligand Interaction Fingerprinting (PLIF) tool of the MOE 2015.10 software was used to analyze the interaction patterns between the ligands with the crucial residues at the binding pocket.

## Molecular dynamics simulations

The Gromacs 2020.2 software [29, 30] was used to assess the stability of the docked complex and to estimate the dynamic behavior of protein–ligand complex to obtain the precise binding modes [31]. The simulations were performed on IL-6 and IL-6Rα, both in the apoprotein form and in complexes with the small molecules. The topology of the proteins was prepared by the pdb2gmx module of GROMACS using the all-atom CHARMM27 force field [32]. Then, the ligands were fully hydrogenated by the Avogadro software [33] before creating topology in the Swissparam web server (http://www.swissparam.ch) [34]. The complexes were solvated in a cubic box and kept at a distance of 10 Å from the edges of the solvated box. Sodium and chloride ions were added to neutralize the charge of the system, which would then be energy minimized using the steepest descent algorithm. In all simulations, the reference temperature was 300 K for the NVT (isothermal-isochoric) ensemble, and the reference pressure was 1 atm for the subsequent NPT (isothermal-isobaric) ensemble [35]. The trajectories were saved every 0.01 ns. Molecular dynamics results were used to calculate the root mean square deviation (RSMD), the root mean square fluctuation (RMSF) and to evaluate the interactions between the ligands and crucial residues. Particularly, to determine the interaction ability of ligand with the key residues, the occupation of hydrogen bonds formation was analyzed by the VMD software [36]. The two limiting factors were adopted as follows: a distance between hydrogen donor (D) and acceptor (A) atoms of <3.5 Å and an angle D–H. . .A of >120˚ [37]. Besides the hydrogen bond formed between protein and ligands, there were hydrophobic interactions and salt bridges. In this study, the percentage occupancy of a type of contact between a ligand

with residue was calculated by the formula:

$$\% \, Occupancy_i = \left[ \frac{1}{N} \sum_{x=1}^{N} (c_i(t_x)) \right] \times 100\% \tag{1}$$

where $N$ is the number of frames of the trajectory, $c_i$ is the total number of bonds of residue $i$ with the ligand in frame $x$. The frequency values of residues can reach over 100% as they have formed multiple contacts with the ligand [38].

## MM-GBSA binding energy

The gmx_MMGBSA package was applied for free energy calculations based on the single trajectory of GROMACS with CHARMM-27 forcefield [39]. This tool allows free energy calculations using MM/PBSA or GBSA (Molecular Mechanics/ Poisson-Boltzmann or Generalized Born Surface Area) methods. The MM/PB(GB)SA binding free energies of the protein with ligand in solvent can be expressed as [40]:

$$\Delta G_{bind} = \Delta G_{complex} - (\Delta G_{protein} + \Delta G_{ligand}) \tag{2}$$

The free energy of protein, ligand or complex component can be calculated as follows:

$$\Delta G_{bind} = \Delta E_{MM} + \Delta G_{solv} - T\Delta S \tag{3}$$

in which :
$$\Delta E_{MM} = \Delta E_{bond+} \Delta E_{vdW} + \Delta E_{elec} \tag{4}$$

$$\Delta G_{solv} = \Delta G_{PB/GB} + \Delta G_{SA} \tag{5}$$

In the above equations, $\Delta E_{MM}$, $\Delta G_{solv}$, and $T\Delta S$ are the changes in the gas phase molecular mechanics energy, solvation free energy, and conformational entropy upon ligand binding. $\Delta E_{bond}$ is the energy of bonded interactions calculated as zero in a dynamic simulation, $\Delta E_{vdW}$ and $\Delta E_{elec}$ are the van der Waals and the electrostatic interactions energy, respectively. $\Delta G_{solv}$ are calculated from polar $\Delta G_{PB/GB}$ (electrostatic solvation energy) and nonpolar $\Delta G_{SA}$ between the solute and the continuum solvent. The polar contribution was estimated using GB-OBC1 model [41, 42], while the nonpolar energy is usually estimated using the solvent-accessible surface area (SASA). In this work, the snapshots sampled from the MD trajectory of each protein-ligand complex were used to carry out the binding free energy calculation using the MM/GBSA approach because of its faster and less computational resources consumption [40]. The entropy ($\Delta S$) effect is minimal on the total energy in the case of comparing the binding states of the ligands with the same protein; hence it can be neglected [43].

## Results and discussions

### 3D-Pharmacophore models

From the PPIs of two proteins, IL-6/IL-6Rα at binding site 1, the 3D-pharmacophore models were constructed by mimicking the properties of IL-6 and IL-6Rα to search for small molecular structures that compete against IL-6 for binding to the IL-6Rα receptor and vice versa [44]. The models Ph_1 and Ph_2 were built based on structures of IL-6 receptor and IL-6, respectively.

In the first model, Ph_1, the selection of pharmacophore points exactly mimicked the components of the receptor IL-6Rα. This model includes 5 features: Phe229 (F3: Aro), Tyr230 (F4: Aro), Phe279 (F5: Hyd), Glu277 (F1: Ani) and Glu278 (F2: Ani). However, small molecules

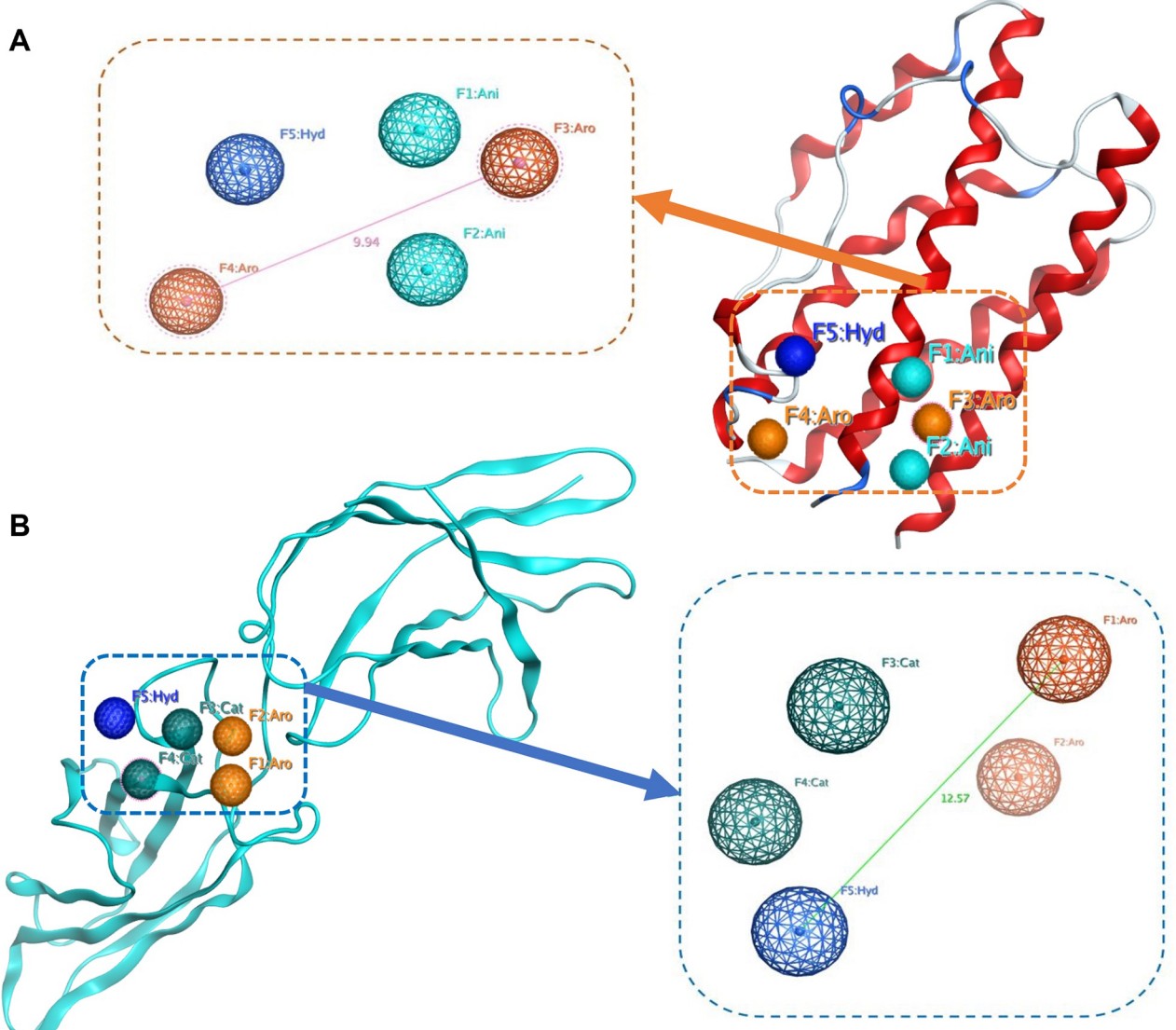

**Fig 2. The pharmacophore models built based on PPI approach are mapped on IL-6 (Ph_1, A) and IL-6Rα (Ph_2, B) structures.** Each sphere represents an abbreviated pharmacophore feature: Cat is cation atom (in dark green), Ani is anion atom (in cyan), Hyd is hydrophobic (in dark blue) and Aro is an aromatic ring (in orange). The pink and green lines represent the largest distance (Å) between pharmacophore points.

that are suitable to become oral drug candidates are often difficult to satisfy a pharmacophore model with too many charged (ani/cat) and aromatic ring points. Therefore, at least one point of either F1 or F2 and F3 or F4 was constrained, respectively; F5 was an essential point in the model. The Ph_1 model and its alignment to the IL-6 backbone are illustrated in Fig 2A.

The second model, Ph_2, revealed five pharmacophore features, including Phe74 (F1: Aro), Phe78 (F2: Aro), Arg179 (F3: Cat), Arg182 (F4: Cat), and Leu178 (F5: Hyd), which were formed based on hot-spots residues of IL-6. Through interactions with the key amino acids of IL-6Rα, similar to Ph_1, this model must satisfy at least one point of either F1 or F2 and F3 or F4 was constrained, respectively; F5 was an essential point. The results of the Ph_2 model and its alignment to the IL-6R are illustrated in Fig 2B.

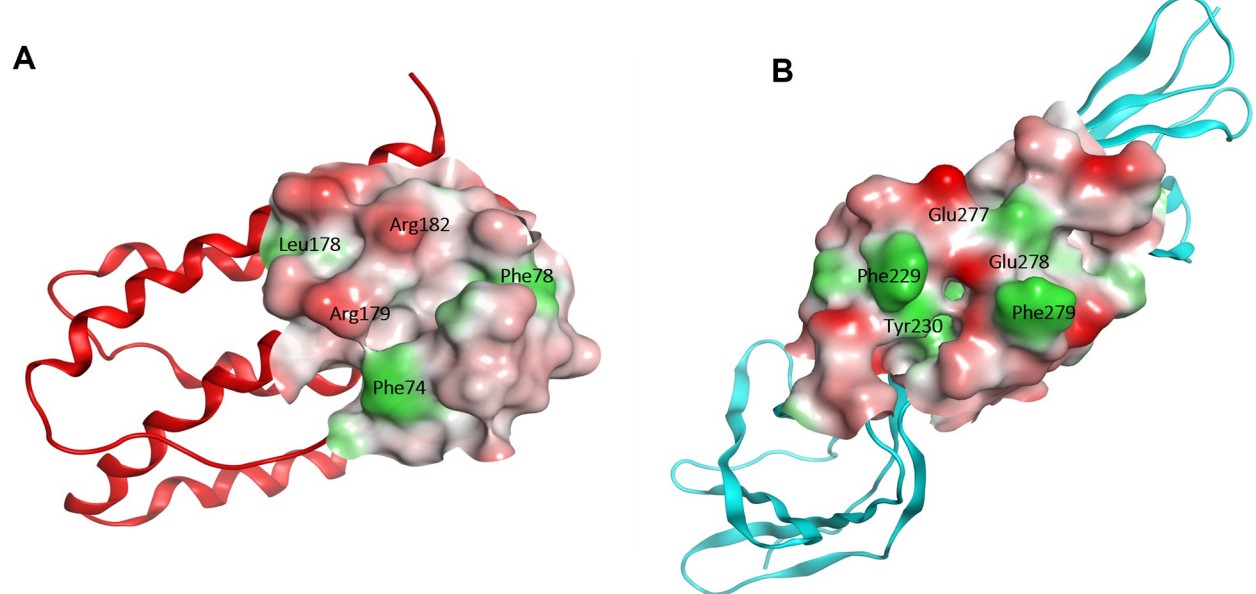

**Fig 3. The molecular docking models D- IL-6 (A) and D-IL-6Rα (B) were generated by the hot-spot residues at the binding site.**

## Molecular docking and in silico screening

The structures of IL-6 and its receptor were prepared by MOE 2015.10 and loaded into the LeadIT 2.1.8 software. At the active site, key residues including Phe74, Phe78, Leu178, Arg179, and Arg182 of IL-6 or Phe229, Tyr230 Glu277, Glu278 and Phe279 of IL-6Rα were selected. Two molecular docking models are named D-IL6 (Fig 3A) and D-IL-6Rα (Fig 3B).

Inspection of the D-IL-6 model reveals a relatively deep binding cavity with a rough and undulating surface, in which significant residues Phe74, Arg179 and Arg182 are located at the outer convex corners, while Phe78 and Leu178 are recessed into the cavity (Fig 3A). It was predicted that the ligands would more readily interact with outward residues than introverted ones. The D-IL-6Rα model is different from the D-IL-6 in that it resembles a trench rather than a pocket, which can serve as a perfect binding site for ligands. In detail, the ligands can readily interact with Phe229, Phe279 and Glu278 on the cavity surface, and to a lesser extent with Glu277 and Tyr230 located inside (Fig 3B).

## In silico screening

The two models, Ph_1 and Ph_2, were uploaded to ZINCPharmer to rapidly screen through the ZINC12 library for compounds with appropriate properties to bind the IL-6 and its receptor. A database of 21,777,093 compounds with more than 200 million conformations was virtually screened through the two pharmacophore hypotheses. Through the pharmacophore models Ph_1 and Ph_2 with a 'druglike' filter to eliminate those violating the Lipinski's "Rule of Five", the screening resulted in 29,677 and 4,481 compounds satisfying the Ph_1 and the Ph_2 model, respectively. All substances that met the two models were then filtered through the ADMET Predictor 10.0 software to search for candidates suitable for oral use with pharmacokinetic profiles likely to pass preclinical trials. The compounds selected by ADMET prediction were then docked into the binding pocket. As a result, 6,893 and 675 substances were successfully docked into the D_IL6 and the D_IL6Rα model, respectively.

**Table 2. The docking scores of top 12 ligands and their interactions with IL-6/IL-6Rα.**

| Rank | Ligand ID | Docking Score (kJ/mol) | IL-6/ IL-6Rα residues interaction | |
|------|-----------|------------------------|------------------------------------|--|
| 1 | ZINC04256801 | −27.57 | **IL-6 residues interaction** | Phe78, Leu178, Arg179, Arg182 |
| 2 | ZINC00753055 | −27.48 | | Phe74, Arg179, Arg182 |
| 3 | ZINC20247718 | −27.06 | | Phe78, Arg179, Arg182 |
| 4 | ZINC02997430 | −25.64 | | Phe74, Arg179, Arg182 |
| 5 | ZINC03000225 | −25.43 | | Phe74, Arg179, Arrg182 |
| 6 | ZINC59449112 | −25.20 | | Phe74, Phe78, Arg179, Arg182 |
| .7 | ZINC02682855 | −25.00 | | Phe74, Arg179, Arg182 |
| 8 | ZINC32853685 | −26.44 | **IL-6Rα residues interaction** | Phe229, Glu277, Glu278, Phe279 |
| 9 | ZINC57774399 | −26.24 | | Phe229, Tyr230, Glu277, Glu278, Phe279 |
| 10 | ZINC72026870 | −25.31 | | Phe229, Tyr230, Glu277, Glu278 |
| 11 | ZINC46227820 | −25.25 | | Phe229, Tyr230, Glu277, Glu278 |
| 12 | ZINC83804241 | −25.07 | | Phe229, Tyr230, Glu277, Glu278 |

In this study, the two pharmacophore models were constructed based on the hot-spot residues of the interaction site between IL-6 and IL-6Rα; fingerprints analysis of protein-ligand interaction showed that most of the ligands could interact with these key residues. In particular, Phe74, Agr179 and Arg182 of IL-6 formed interaction with ligands at the high ratio of 81%, 97% and 58%, respectively (S1A Fig). Similarly, at the binding site of IL-6Rα, Glu278, Glu277 and Phe229 formed interaction with a significant proportion of ligands (96%, 42%, and 56%, respectively) (S1B Fig). These results proved that 3D-pharmacophore models could accelerate the identification of compounds with suitable physicochemical properties for binding to the IL-6 and its receptor from the diverse database as ZINC. Besides, the high percentage of successfully docked ligands suggested the effectiveness and reliability of these models.

Substances with good docking scores < −20kJ/mol were predicted to strongly bind to the protein target. The results of 235 compounds were obtained, accounting for 31.05% of all the docked ligands and listed in the S1 Table. Among them, the top 12 ligands that satisfied Lipinski's rule had docking scores < −25 kJ/mol, and were bound to at least three critical residues of the protein targets were selected for further investigation using MD simulations and the occupancy of the hydrogen bond (Table 2, S2 Table and S2 Fig).

## Molecular dynamics simulation

Firstly, the MD trajectories of IL-6 and its receptor either in the form of apoprotein or complex with 12 ligands were investigated during a simulation time of 50 ns to analyze the stability of these proteins, ligands, and their interactions. Comparison of MD trajectories between apoprotein and complex was formed to simultaneously evaluate protein fluctuation (RMSD), and the impact of ligand binding on protein residues (RMSF) in these two states. As we know, many recent studies, especially those involving MD simulation in SAR-CoV2 inhibitor studies, have applied similar comparative analysis [45, 46].

**The stability of the IL-6/IL-6Rα and ligands.** The RMSD values of IL-6/IL-6R and complexes were investigated to evaluate the stability of each respective structure during molecular dynamics simulation. In general, all complexes of IL-6/IL-6Rα with 12 ligands had RMSD values lower than that of the apoprotein (Apoprotein-IL6: 2.62 ± 0.29 Å, Apoprotein-IL6Rα: 2.13 ± 0.72 Å) except for ZINC32853685 (Table 3). As can be observed from the time plot of the RMSD values of the carbon backbone, all 12 complexes fluctuate less than 2 Å (S3 and S4 Figs). For the apoprotein-IL-6, although the RMSD values strongly deviated in the initial 20 ns, it became stable after that till the end of the simulation time. Remarkably, the highly stable

**Table 3. The mean and standard deviation of protein RMSD, ligand RMSD and protein RMSF of complexes between IL-6/IL-6Rα and 12 selected ligands.**

| Complex | RMSD of protein C$_{backbone}$ (Å) | RMSD of heavy atoms of ligand (Å) | RMSF of carbon alpha (Å) |
|---|---|---|---|
| Apoprotein-IL6 | 2.62 ± 0.29 | | 1.06 ± 0.68 |
| IL-6-ZINC04256801 | 1.95 ± 0.25 | 0.98 ± 0.39 | 1.09 ± 0.48 |
| IL-6-ZINC00753055 | 2.29 ± 0.21 | 1.42 ± 0.49 | 1.00 ± 0.53 |
| IL-6-ZINC20247718 | 1.72 ± 0.23 | 2.51 ± 0.59 | 1.04 ± 0.58 |
| IL-6-ZINC02997430 | 2.16 ± 0.37 | 0.68 ± 0.19 | 0.97 ± 0.61 |
| IL-6-ZINC03000225 | 2.36 ± 0.32 | 1.94 ± 0.35 | 0.99 ± 0.61 |
| IL-6-ZINC59449112 | 2.11 ± 0.31 | 2.23 ± 0.39 | 1.08 ± 0.66 |
| IL-6-ZINC02682855 | 1.90 ± 0.19 | 1.68 ± 0.30 | 0.89 ± 0.39 |
| Apoprotein-IL6Rα | 2.13 ± 0.72 | | 1.54 ± 0.64 |
| IL-6Rα-ZINC32853685 | 3.10 ± 0.82 | 3.10 ± 0.56 | 1.60 ± 0.80 |
| IL6-Rα-ZINC57774399 | 1.94 ± 0.47 | 2.16 ± 0.50 | 1.38 ± 0.62 |
| IL6-Rα-ZINC72026870 | 1.69 ± 0.40 | 1.80 ± 0.17 | 1.35 ± 0.53 |
| IL6-Rα-ZINC46227820 | 1.96 ± 0.49 | 1.98 ± 0.49 | 1.34 ± 0.55 |
| IL6-Rα-ZINC83804241 | 1.80 ± 0.43 | 1.93 ± 0.29 | 1.26 ± 0.56 |

systems (RMSD ≤1 Å) were the complexes of IL-6 with ZINC04256801, ZINC00753055, and ZINC20247718 with equilibrium states reached after 2, 5, and 10 ns, respectively (S3 Fig). On the other hand, large deviations could be seen for the apoprotein-IL6-Rα and most of their complexes (RMSD from 1 to 3 Å) (S4 Fig).

To identify highly stable ligands during 50 ns MD simulations, the RMSD obtained from protein fitting its ligands were plotted and analyzed (Table 3, S3 and S4 Figs). High RMSD values revealed large conformational changes of the ligands during MD simulations. Most of the ligands had mean RMSD values <2 Å except for ZINC20247718, ZINC59449112, ZINC32853685, ZINC57774399. Notably, ZINC02997430, ZINC03000225, ZINC72026870 and ZINC83804241 reached stability with relatively low RMSD fluctuation only about 1 Å after 3, 4, 8, and 10 ns, respectively. The data suggest that the majority of both proteins and ligands were stable in their complexes throughout the simulation.

**The mobility of key residues in IL-6/IL-6Rα.** The root mean square fluctuation (RMSF) is necessary to characterize the local conformational change in the protein chain and the ligands. The RMSF profiles of the apoprotein and the complexes calculated by residues index Cα were similar during the dynamics simulations (Table 3). However, subtle differences were observed for a few regions, including the loop at residues Ala61–Asp71 of IL-6. Interestingly, the critical residues at the binding site such as Phe74, Phe78, Leu178, Arg179 and Arg182 of IL-6 (S5A Fig) and Phe229, Tyr230, Glu277, Glu278 and Phe229 of IL-6Rα (S5B Fig) had stable fluctuations with RMSF <2 Å for the whole 50 ns simulation. Therefore, the results strongly suggested that protein-ligand complexes were stable at their binding site.

**Hydrogen bond occupancy by the active binding residues.** Hydrogen bonds (H-bonds) play a significant role in ligand binding. The H-bond occupancies were calculated from the trajectories of 50 ns MD simulations. Six ligands with the H-bond occupation of >75% were considered strong hydrogen bonds [37] (Table 4). ZINC83804241, ZINC72026870 and ZINC46227820 interacted with key residues of IL-6Rα by hydrogen bonding at very high occupation. Among them, ZINC83804241 had the highest interaction frequency. This compound formed strong hydrogen bonds with two hot-spot residues: Glu278 (492.14%) and Phe229 (92.34%). ZINC72026870 and ZINC46227820 acted as hydrogen donors to Glu278 and Phe229. In particular, ZINC72026870 had a percentage occupation of 106.78% with Glu278 and 31.94% with Phe229. For ZINC46227820, the occupied ratio in these two residues

**Table 4. Hydrogen bond occupancy of 6 top hit ligands from data of the 50ns MDs trajectories.**

| Complex | Donor | Acceptor | Occupancy (%) |
|---|---|---|---|
| IL-6-ZINC04256801 | Arg182-Side | LIG185-Side | 96.48 |
| | Arg179-Side | LIG185-Side | 16.78 |
| | Arg30-Side | LIG185-Side | 8.16 |
| | Gln75-Side | LIG185-Side | 4.06 |
| | LIG185-Side | Gln75-Main | 6.16 |
| IL-6-ZINC02997430 | Arg179-Side | LIG185-Side | 171.67 |
| | Arg179-Side | LIG185-Main | 54.84 |
| | Ser176-Side | LIG185-Side | 158.06 |
| | LIG185-Side | Ser176-Side | 38.46 |
| | Gln175-Side | LIG185-Side | 49.90 |
| | LIG185-Side | Cys73-Main | 48.74 |
| IL-6-ZINC03000225 | Arg182-Side | LIG185-Side | 81.67 |
| | Arg179-Side | LIG185-Side | 16.71 |
| | Gln175-Side | LIG185-Side | 12.79 |
| | Gln75-Side | LIG185-Side | 14.57 |
| | LIG185-Side | Gln75-Side | 11.46 |
| | Gln75-Main | LIG185-Side | 10.62 |
| IL6-Rα-ZINC72026870 | LIG299-Side | Glu278-Side | 106.78 |
| | LIG299-Side | Phe229-Main | 31.94 |
| | LYS252-Side | LIG299-Side | 138.38 |
| IL6-Rα-ZINC46227820 | LIG299-Side | Glu278-Side | 103.42 |
| | LIG299-Side | Phe229-Main | 20.95 |
| | LIG299-Side | Asp221-Side | 39.4 |
| IL6-Rα-ZINC83804241 | LIG299-Main | Glu278-Side | 392.92 |
| | LIG299-Side | Glu278-Side | 99.22 |
| | LIG299-Side | Phe229-Main | 92.34 |

was 103.42% and 20.95%, respectively. For the IL-6 target, ZINC02997430 expressed strong inhibitory potential on this receptor with the highest hydrogen bonds occupancy to the hot-spot residue Arg179 (226.51%). ZINC04256801 and ZINC03000225 served as hydrogen acceptors from Arg182 and Arg179. Among them, ZINC04256801 formed a total frequency of 96.48% for Arg182 and 16.78% for Arg179. At the same time, the interactive percentages of these two amino acids with ZINC03000225 were 81.67% and 16.71%, respectively. The hydrogen bond occupancy is the key to evaluating the interaction stability of protein-ligand complexes. Thence, in the next step, 100 ns MD simulations were performed and analyzed for these six top ligands to closely assess their receptor-binding capacity.

## Identification of potential inhibitors by long-time scale MD simulations

Based on analysis of RMSD, RMSF, and hydrogen bond interaction with key residues, the complexes of ZINC04256801, ZINC03000225, ZINC02997430 with IL-6 and the complexes of ZINC83804241, ZINC72026870, ZINC46227820 with IL-6Rα were selected to perform a longer MD simulation of 100 ns. Similar to 50 ns MD simulations, the results of 100 ns simulations were evaluated based on three following criteria.

For the IL-6 target, Fig 4A showed the backbone RMSD curves for IL-6 and its complexes with ZINC04256801, ZINC03000225, ZINC02997430. The protein complex with ZINC02997430 reached equilibrium after about 5 ns with a considerably low RMSD

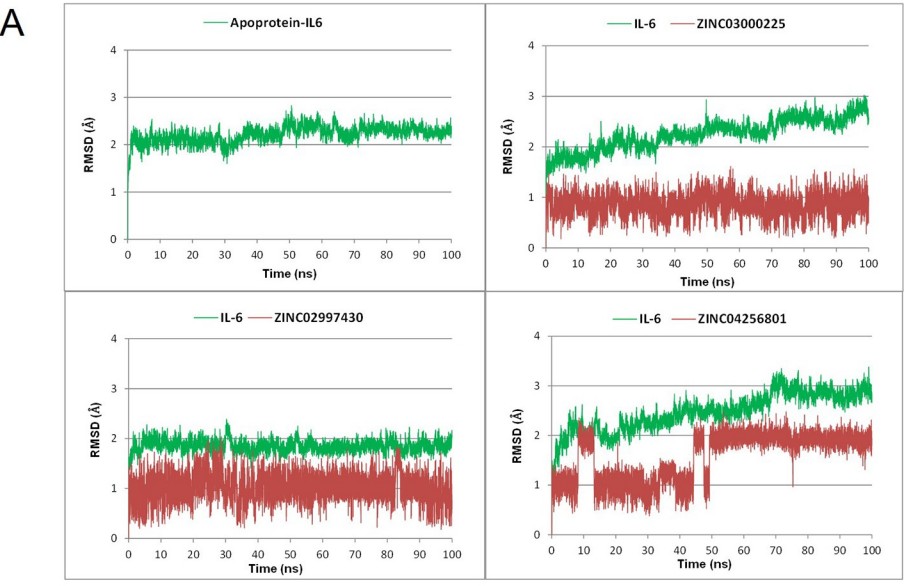

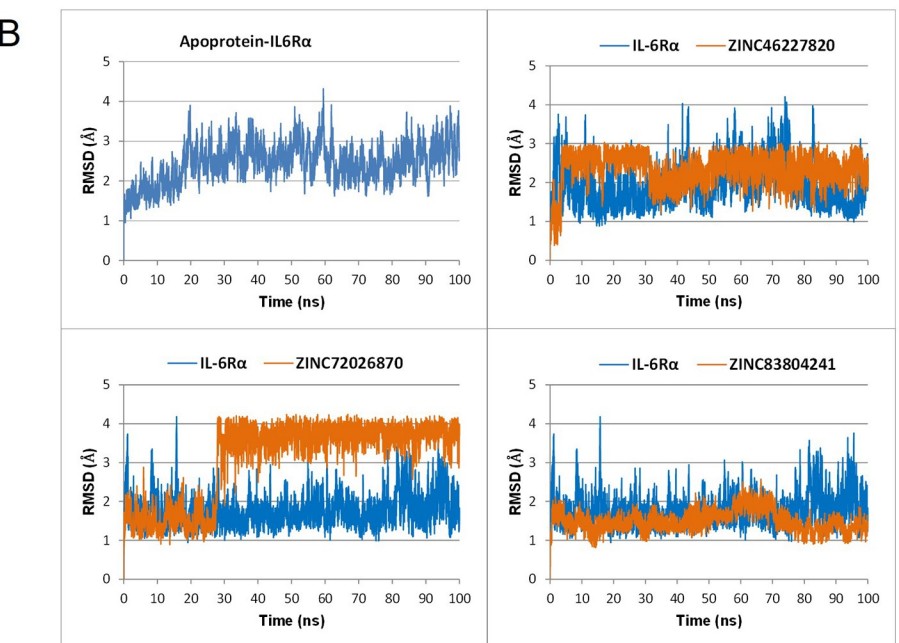

**Fig 4. Carbon backbone RMSD profiles of the IL-6 (A) and IL-6Rα (B) in apoprotein form and in complexes with 6 top ligands.**

fluctuation of ≤0.5 Å. Meanwhile, the complexes of IL-6 with the other 2 ligands showed high volatility during the MD simulation with RMSD fluctuation of about 1 Å (from 2–3 Å). These results were in full agreement with the RMSF value observed on the graph (Fig 5A). Some atoms in these two complexes had high RMSF values, rendering RMSD distance fluctuated. However, when comparing the ligand's RMSD values, ZINC03000225 was more stable than ZINC02997430 and ZINC04256801 throughout the simulation (Fig 4A). To have a closer look at the binding modes, frames were extracted at every 25 ns from the last 50 ns trajectories and the ligands rendered at the binding cavity (S6 Fig). Out of the three ligands, ZINC02997430

**A**

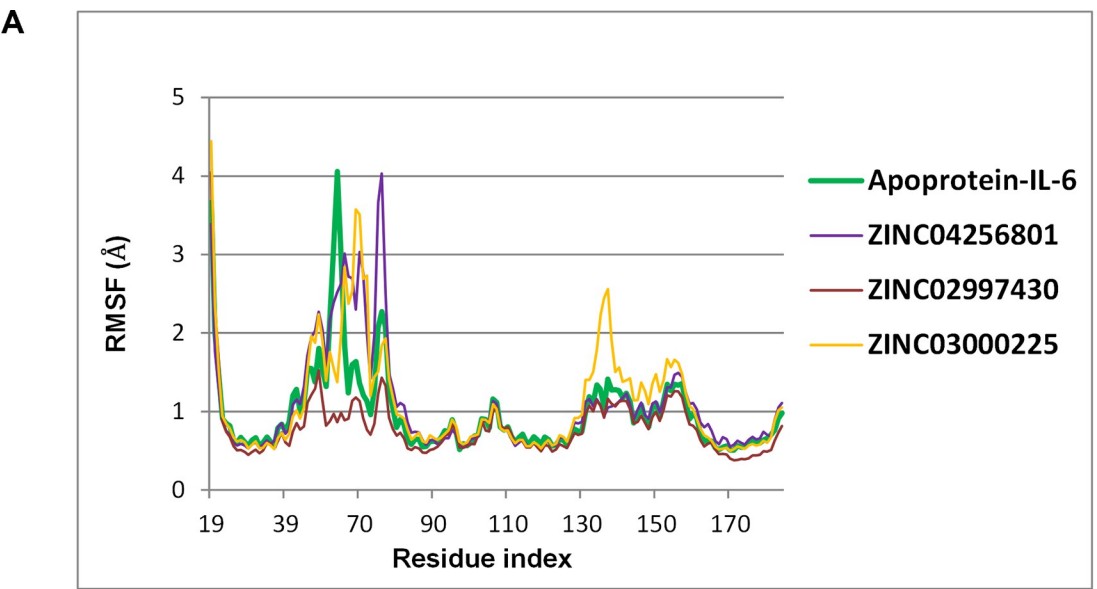

**B**

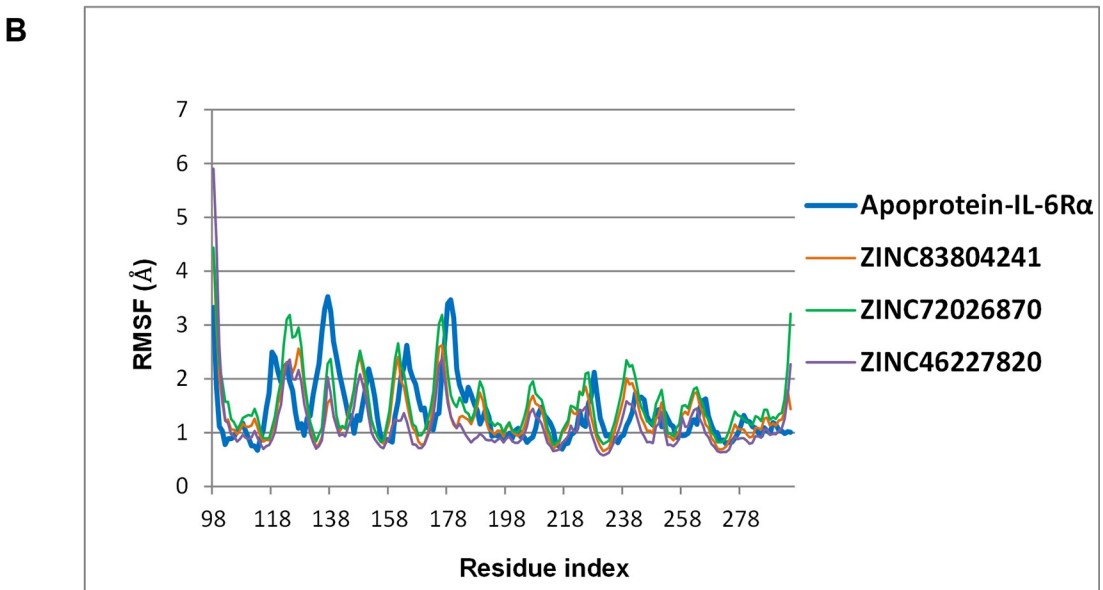

**Fig 5. Carbon alpha RMSF values of the apoprotein IL-6 (A) and IL-6Rα (B) its complexes with the 6 top ligands.**

could maintain the contacts with IL-6 during the MD simulation from 50 ns to 100 ns. In comparison, ZINC04256801 and ZINC03000225 seemed to move out of the binding cavity from the last 75 ns simulation time and formed unstable complexes.

The stability of the complexes with protein for the IL-6R target was illustrated in Fig 4B. Throughout the simulations, all systems were relatively stable with RMSD values deviating <2 Å. The RMSF of the Cα atom of each residue of the apoprotein IL-6Rα and its complexes were similar (Fig 5B). The RMSD value of ligands' heavy atoms (Fig 4B), indicated that the ligands reached high stability after about 30 ns. In particular, ZINC72026870 and ZINC83804241 had RMSD values of <1 Å. Extractions of frames at every 25 ns from the last 50 ns trajectories and

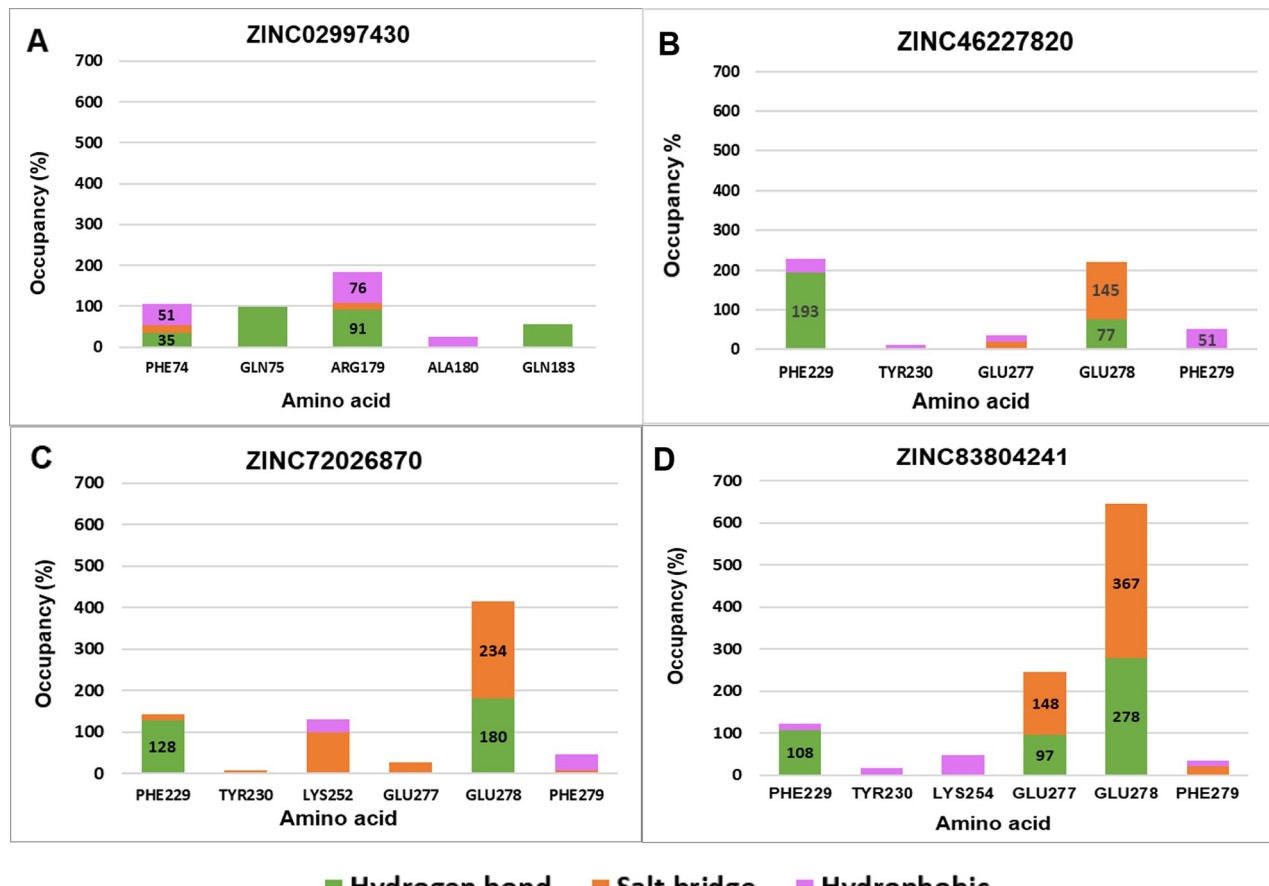

**Fig 6. The interactions occupancy between 4 top hit ligands and critical amino acids of IL-6/IL-6Rα.** In which, Fig 6A–6D correspond to ZINC02997430, ZINC46227820, ZINC72026870, and ZINC83804241, respectively. Color between columns distinguishes interaction tones of different binding types.

visualization of the ligands at the binding cavity unambiguously showed that all 3 ligands binding strongly to the IL-6R at the binding pocket during the 100 ns simulation (S7 Fig).

To further validate protein-ligand interactions, a detailed analysis of the hydrogen-bonding (H-bond), salt bridge, and hydrophobic interactions between the ligands and proteins and binding free energy calculation was carried out for 4 top hit compounds. All 4 ligands could maintain stable contacts at the binding cavity during the entire 100 ns MD simulation.

Analysis of interactions occupancy on each ligand binding to IL-6/IL-6Rα showed that all four ligands exhibited strong interactions with the key residues (Fig 6). Some residues had interaction frequency >100% because they formed multiple same interactions simultaneously.

Interestingly, for the IL-6Rα protein, all 3 compounds could dock to 5 hot-spot residues, namely Phe229, Tyr230, Glu277, Glu278, and Phe279. Notably, Phe229 and Glu278 occupied high frequencies of above 100%. In detail, for Glu278, ZINC83804241 had the highest occupancy of 640%, followed by ZINC72026870 and ZINC46227820 with the percentage frequency of over 420% and 220%, respectively. The high occupancies suggest that these ligands have a special affinity for Glu278 in IL-6Rα. This result was consistent with the observation that IL-6Rα mutated at Glu278 completely lost the ability to bind to IL-6 [20]. On the other hand, a statistical study of proportion bonds between ZINC02997430 and IL-6 showed that this ligand only interacted with 2/5 key residues of IL-6, including Arg179 and Phe74. However, the

interative percentage of these two amino acids was relatively high with 185% and 105% for Arg179 and Phe74, respectively. It was in good agreement with a previous study [47] where Arg179 was identified as the most important residue on IL-6 required for IL-6Rα binding.

ZINC02997430 formed both hydrogen bonds and hydrophobic interactions with the two key residues Arg179 and Phe74 of IL-6 with high frequency. At the phenyl ring of the benzoate side branch, the ligand participated in π-alkyl and π-π interactions with Arg179 and Phe74 with a frequency of 76% and 51%, respectively. In addition, the carboxylate group ($-COO^-$) of this branch chain also accepted hydrogen bonds from Phe74 with a frequency of 35%. At the main chain of the molecule, the nitro group ($-N^+OO^-$) formed stronger hydrogen bonds with Arg179 with a frequency 91% (Fig 6A and S8A Fig).

The structural core of molecules ZINC83804241, ZINC72026870, and ZINC46227820 contains a piperazine cyclic consisting of a six-membered ring containing two nitrogen atoms at the opposite. These N atoms ($-N^+$) were the main agents that created the salt bridges with the two key residues Glu277 and Glu278 of IL-6Rα. In particular, the ratio of salt bridge interaction between three above ligands with Glu278 with frequency of 367% (Fig 6D and S8D Fig), 234% (Fig 6C and S8C Fig) and 145% (Fig 6B and S8B Fig), respectively. While the key residues Glu277 only formed a strong salt bridge interaction with ZINC83804241 with frequency of 148% (Fig 6D and S8D Fig). ZINC83804241 can be considered as the strongest salt-bridging ligand with IL-6Rα. Besides the salt bridge interaction, these three ligands also participate in hydrogen bonds and hydrophobic interactions with the key residues Phe229, Tyr230, Glu277, Glu278, and Phe279 of IL-6Rα. Similar to 50 ns MD simulations, the results of 100 ns MD trajectories suggest that ZINC46227820 acted as hydrogen donors to Glu278 and Phe229. The −NH groups of piperazin cyclic and the side chain donated hydrogen bonds with Phe229 and Glu278 with frequency of 193% and 77%, respectively (Fig 6B and S8B Fig). In addition, this compound also interact with Phe279 by π-π interaction at the phenyl ring of the main chain with a frequency of 51% (Fig 6B and S8B Fig). On the other hand, Glu278 and Phe229 also accepted hydrogen bonds from the −NH groups of ZINC83804241, ZINC72026870 with a high frequency of over 100%, but the phenyl groups formed a weak π-π interaction with Tyr230 and Phe279 with a very low occupancy of under 50% (Fig 6C and 6D, S8C and S8D Fig). Similar to salt bridge interaction above, Glu277 only interacted strongly with ZINC83804241 by accepting hydrogen bonds from −NH groups of piperazin ring with a frequency of 97% (Fig 6D and S8D Fig).

**MM-GBSA binding free energy.** Evaluation of the $\Delta G_{bind}$ can help unambiguously identify the most potential receptor inhibitors. However, the aim of this study was to discover the small molecules that inhibit the PPI of the IL-6/IL-6Rα complex by binding strongly and specifically at the location of the key residues on these two proteins. Therefore, the interaction of each ligand with these residues should be carefully considered to identify top hit compounds. The molecular mechanics Generalized Born surface area (MMGBSA) approach efficiently recapitulates the binding capacity of a small molecule to the target. MMGBSA binding free energy simulations were obtained for IL-6/IL-6Rα and its complexes with its top hit ligands calculated from the 100 ns MD simulation (Table 5).

All the 4 ligands had stability until the end of the simulations with the negative binding energies. In particular, the fluctuations of these complexes over time showed that ZINC02997430, ZINC72026870, and ZINC83804241 complexes gave a stable energy value, whereas that of ZINC46227820 fluctuated strongly and had a $\Delta G_{bind}$ value close to 0 kcal/mol from the last 50 ns of MD trajectory (Fig 7). For the IL-6Rα target, ZINC83804241 gave the lowest binding energy averaging at −30.28 kcal/mol and had the highest binding affinity to the protein target. ZINC72026870 and ZINC46227820 have higher $\Delta G_{bind}$ values, indicating the weaker interaction between these compounds in the IL-6Rα active site. Notably,

**Table 5. The calculation of binding free energy results of 4 top hit ligands.**

| Complex | IL-6-ZINC02997430 | IL6-Rα-ZINC72026870 | IL6-Rα-ZINC46227820 | IL6-Rα-ZINC83804241 |
|---|---|---|---|---|
| $\Delta E_{vdW}$ (kcal/mol) | −26.48 ± 3.55 | −14.21 ± 4.47 | −17.14 ± 4.22 | −14.78 ± 4.18 |
| $\Delta E_{elec}$ (kcal/mol) | −28.13 ± 9.28 | −354.25 ± 28.73 | −230.58 ± 35.90 | −411.32 ± 40.02 |
| $\Delta G_{GB}$ (kcal/mol) | 39.10 ± 7.75 | 355.95 ± 25.82 | 242.22 ± 31.68 | 399.67 ± 37.51 |
| $\Delta G_{SA}$ (kcal/mol) | −3.63 ± 0.39 | −3.76 ± 0.47 | −3.06 ± 0.56 | −3.84 ± 0.42 |
| $\Delta G_{gas}$ (kcal/mol) | −54.61 ± 10.08 | −368.46 ± 28.28 | −247.73 ± 34.24 | −426.11 ± 39.81 |
| $\Delta G_{sol}$ (kcal/mol) | 35.46 ± 7.58 | 352.19 ± 25.64 | 239.16 ± 31.41 | −395.83 ± 37.33 |
| $\Delta G_{bind}$ (kcal/mol) | −19.15 ± 4.04 | −16.26 ± 5.13 | −8.57 ± 4.63 | −30.28 ± 5.14 |

ZINC83804241 and ZINC72016870 binding energy tended to decrease at the end of the simulation and reflected an increase in binding affinity. With good binding energy in the range of −19.15 kcal/mol, ZINC02997430 exhibited a relatively strong binding affinity for IL-6.

Finally, based on analyses including stability of complexes and ligands, occupancy frequencies between effects, MM/GBSA binding free energies, ZINC83804241 and ZINC02997430 were identified as the most potential compounds. Fig 8 clearly demonstrated the interaction between ZINC02997430 and ZINC83804241 with the crucial residues at the active sites of IL-6 and IL-6Rα, respectively.

These results strongly suggested the promising lead compounds for the design of novel IL-6 inhibitors by blocking the PPI of IL-6 and IL-6Rα. In addtion, in recent studies looking for IL-6 inhibitors to acute respiratory distress syndrome in severe covid-19 patients, monoclonal antibodies such as tocilizumab, sarilumab, siltuximab emerged as a top option. These targeted monoclonal antibodies can also reduce downstream IL-6 signaling pathways through direct inhibition of the interaction between IL-6 with IL-6Rα at site I [48]. Therefore, it can be seen that the search for small molecule drugs capable of inhibiting site I is a potential research direction.

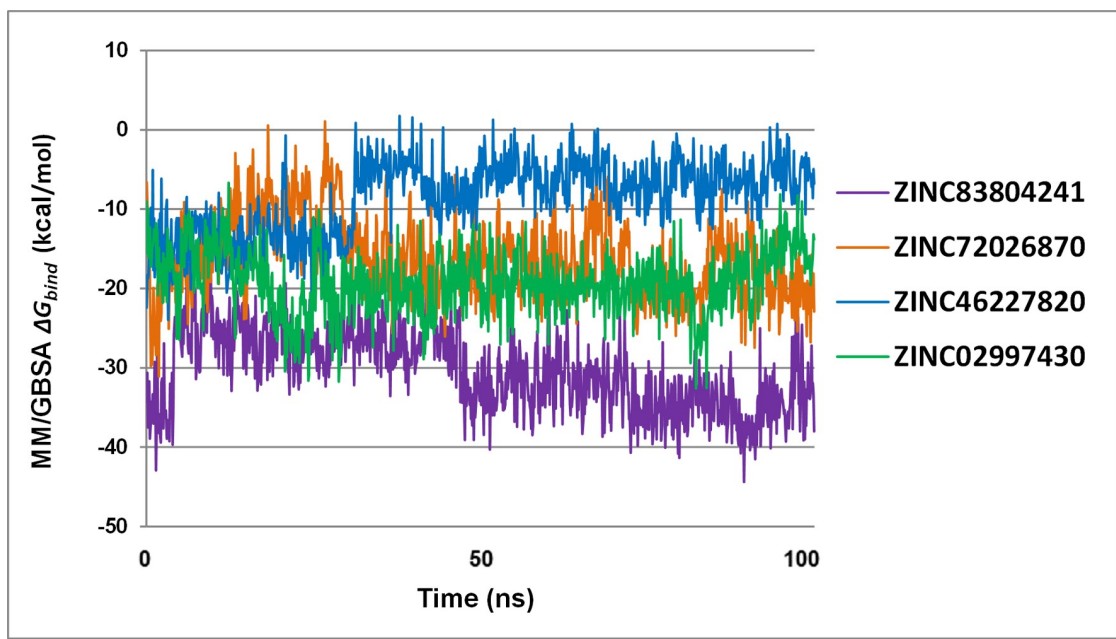

**Fig 7. Total energy binding value for the 100 ns MD simulation of 4 top hit ligands.** Including ZINC02997430 (in green), ZINC83804241 (in purple), ZINC72026870 (in orange) and ZINC46227820 (in blue) as per calculated using the MMGBSA method.

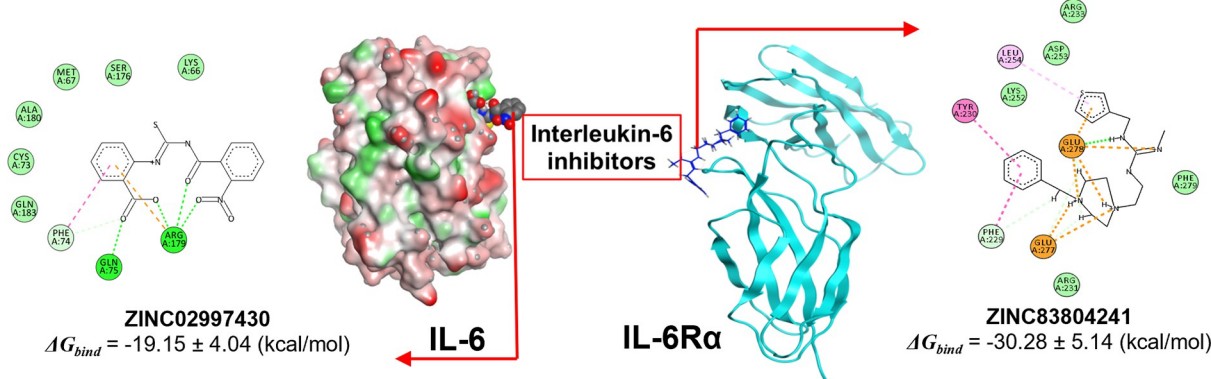

**Fig 8. The interaction between ZINC02997430 and ZINC83804241 with the crucial residues at the active site of IL-6 and IL-6Rα, respectively.**

## Conclusions

In this study, we developed structure-based pharmacophore models and performed molecular dock-ing and MD simulation to search for substances capable of inhibiting IL-6. These approaches are considered high-performance in silico screening methods, capable of quickly searching for compounds from large databases, thus saving time and significantly reducing research costs. IL-6 is a potential target for anti-inflammatory drug discovery, especially in the present period when the world is heavily affected by the COVID-19 pandemic. The current work screened for small molecules which inhibit IL-6 or IL-6 receptor (IL-6Rα) in the hope of finding a potential therapeutic agent. From the approximately 22 million substances of ZINC12, the study constructed and screened using pharmacophore models, assessed by ADMET prediction and docking. Subsequent application of MD simulation was conducted for 12 compounds predicted to bind strongly to IL-6/IL-6Rα. Detailed analysis based on the occupation percentage of hydrogen bonding, hydrophobic, salt bridge in combination with binding free energy calculation allowed us to identify ZINC83804241 and ZINC02997430 as the most potential IL-6 inhibitors. Results from the present study suggested that these potential compounds can be developed into novel IL-6 inhibitors.

## Supporting information

**S1 Fig. PLIF analysis of the docking process.** The interaction frequency of individual residue on IL-6 (A) and IL-6Rα (B) with the docking poses of ligands.
(TIF)

**S2 Fig. Interactions between 12 top ligands and IL-6/IL-6Rα.** The yellow and blue ligands are potential compounds binding to IL-6 and IL-6Rα, respectively.
(TIF)

**S3 Fig. Carbon backbone RMSD profiles of IL-6 in apoprotein form and in complexes with the 7 ligands and heavy atom RMSD profiles of the respective ligands calculated by 50 ns MDs trajectories.**
(TIF)

**S4 Fig. Carbon backbone RMSD profiles of IL-6Rα in apoprotein form and in complexes with the 5 ligands and heavy atom RMSD profiles of the respective ligands calculated by 50

**ns MDs trajectories.**
(TIF)

**S5 Fig. Carbon alpha RMSF values of the apoprotein IL-6/IL-6Rα and its complexes with the 12 ligands.** S4A Fig illustrated the RMSF Cα values of the IL-6 apoprotein and its complexes with 7 ligands, S4B Fig presented RMSF Cα values of the IL-6Rα apoprotein and its complexes with 5 ligands calculated by 50 ns MDs trajectories.
(TIF)

**S6 Fig. Protein-ligand binding modes in MD simulations of top hit ligands of IL-6.** Protein-ligand conformations at every 25 ns of the last 50 ns MDs trajectories for ZINC03000225, ZINC04256801, ZINC02997430.
(TIF)

**S7 Fig. Protein-ligand binding modes in MD simulations of top hit ligands of IL-6Rα.** Protein-ligand conformations at every 25 ns of the last 50 ns MDs trajectories for ZINC46227820, ZINC72026870, ZINC83304241.
(TIF)

**S8 Fig. The detailed interaction diagram of the ligand atom with the IL-6 and IL-6Rα residues.** S8A-S8D Fig correspond to ZINC02997430, ZINC46227820, ZINC72026870, and ZINC83804241, respectively.
(TIF)

**S1 Table. The docking results of 235 potential IL-6 inhibitors.**
(PDF)

**S2 Table. The chemical structures and properties of potential IL-6/IL-6Rα binders.**
(PDF)

## Acknowledgments

The authors would like to thank Mr. Dinh-Long-Hung Pham for critical reading of the manuscript.

## Author Contributions

**Conceptualization:** Khac-Minh Thai.

**Data curation:** Que-Huong Tran, Quoc-Thai Nguyen, Minh-Tri Le.

**Formal analysis:** Que-Huong Tran, Nguyen-Quynh-Huong Vo, Thi-Thuy-Nga Tran, Minh-Tri Le.

**Funding acquisition:** Quoc-Thai Nguyen, Tan Thanh Mai, Minh-Tri Le, Dieu-Thuong Thi Trinh, Khac-Minh Thai.

**Investigation:** Que-Huong Tran, Nguyen-Quynh-Huong Vo, Tan Thanh Mai, Thi-Thuy-Nga Tran.

**Methodology:** Thanh-Dao Tran, Dieu-Thuong Thi Trinh, Khac-Minh Thai.

**Project administration:** Quoc-Thai Nguyen, Minh-Tri Le.

**Resources:** Tan Thanh Mai, Thanh-Dao Tran, Minh-Tri Le, Dieu-Thuong Thi Trinh, Khac-Minh Thai.

**Software:** Nguyen-Quynh-Huong Vo, Dieu-Thuong Thi Trinh, Khac-Minh Thai.

**Supervision:** Quoc-Thai Nguyen, Thanh-Dao Tran, Dieu-Thuong Thi Trinh, Khac-Minh Thai.

**Validation:** Que-Huong Tran, Thi-Thuy-Nga Tran, Thanh-Dao Tran, Dieu-Thuong Thi Trinh.

**Visualization:** Que-Huong Tran, Nguyen-Quynh-Huong Vo, Tan Thanh Mai, Thi-Thuy-Nga Tran, Dieu-Thuong Thi Trinh.

**Writing – original draft:** Que-Huong Tran, Quoc-Thai Nguyen, Nguyen-Quynh-Huong Vo, Tan Thanh Mai, Thi-Thuy-Nga Tran, Thanh-Dao Tran, Minh-Tri Le, Dieu-Thuong Thi Trinh, Khac-Minh Thai.

**Writing – review & editing:** Que-Huong Tran, Quoc-Thai Nguyen, Dieu-Thuong Thi Trinh, Khac-Minh Thai.

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
