## [Decision Letter · Decision Letter 0]

2 Feb 2022

PONE-D-21-38021Structure-Based 3D-Pharmacophore Modeling to Discover Novel Interleukin 6 Inhibitors: An In silico Screening, Molecular Dynamics Simulations and Binding Free Energy Calculations.PLOS ONE

Dear Dr. Thai,

Thank you for submitting your manuscript to PLOS ONE. After careful consideration, we feel that it has merit but does not fully meet PLOS ONE’s publication criteria as it currently stands. Therefore, we invite you to submit a revised version of the manuscript that addresses the points raised during the review process.

We look forward to receiving your revised manuscript.

Kind regards,

Chandrabose Selvaraj, Ph.D.

Academic Editor

PLOS ONE

Journal Requirements:

"This work was supported by the VietNam National Foundation for Science and Technology Development (NAFOSTED) under the Grant Number 108.05-2018.15 (to Quoc-Thai Nguyen). The authors would like to thank Mr. Dinh-Long-Hung Pham for critical reading of the manuscript."

We note that you have provided funding information. However, funding information should not appear in the Acknowledgments section or other areas of your manuscript. We will only publish funding information present in the Funding Statement section of the online submission form. 

"This work was supported by the VietNam National Foundation for Science and Technology Development (NAFOSTED) under the Grant Number 108.05-2018.15 (to Quoc-Thai Nguyen) [https://nafosted.gov.vn/]. Tan Thanh Mai was funded by Vingroup Joint Stock Company and supported by the Domestic Master/ PhD Scholarship Programme of Vingroup Innovation Foundation (VINIF), Vingroup Big Data Insti-tute (VINBIGDATA), code VINIF.2020.TS.128 [https://vinbigdata.org/quy-vinif/]. 

The funders had no role in study design, data collection and analysis, decision to publish, or preparation of the manuscript"

Reviewers' comments:

Reviewer's Responses to Questions

**Comments to the Author**

1. Is the manuscript technically sound, and do the data support the conclusions?

Reviewer #1: Partly

Reviewer #2: Partly

2. Has the statistical analysis been performed appropriately and rigorously? 

Reviewer #1: N/A

Reviewer #2: Yes

3. Have the authors made all data underlying the findings in their manuscript fully available?

Reviewer #1: Yes

Reviewer #2: Yes

4. Is the manuscript presented in an intelligible fashion and written in standard English?

Reviewer #1: Yes

Reviewer #2: No

5. Review Comments to the Author

Reviewer #1: Authors in the present study have highlighted the importance of interleukin in several biological functions such as immune regulation, hematopoiesis, and inflammation. The pathogenic role of IL-6 during ARDS and COVID19 were highlighted. Therefore, the author has selected IL-6 as potential target and has screened for inhibitors using structure based 3D pharmacophore modeling, MD simulation and validation through binding free energy calculation. However few questions need to be addressed and therefore I recommend a major revision of the manuscript to be suitable for publications in PLOS One.

Abstract:

1. Provide space “ membranebound”

2. Check spelling “ inteaction”

3. Revise the sentence “ However, to date, no orally available small-molecule drug has been approved“

4. Remove “however”.

5. Mention the key residues in ()

6. What is the “12 potential substances” here ligands or compounds? Please modify the sentence.

7. Authors should highlight the future studies or importance of the ligands identified as lead in the one sentence in the end of the abstract.

Introduction:

8. Update the number of COVID cases according to the reference [7].

9. Modify the sentence” IL-6 activates cells by binding to IL-6 receptor (IL-6Rα) and gp130. Ternary complex forms 14 a hexamer containing IL-6, IL6Rα, and gp130, sequential and cooperative assembly”. Flaws in continuation of the sentence should be corrected.

10. What does the author mean by “biologics”. Correct it.

11. “Most previous studies mainly focused on '' please correct it. Author has to improve the grammar throughout the manuscript

Materials and methods:

12. Why has the author selected “5FUC.pdb for their studies?

13.Why the author has studied “ adverse effects on genital organs, teratogenicity and developmental were also evaluated through the parameter Reproductive Toxicity” how this is correlated with the present study?

14.The of version of the LeadIT in line 12 can be incorporated in line 7

Results:

15.How the author confirmed the identified molecule is a small molecule? Just rule of five will determine the identified compound as lead? What criteria under rule of five were considered? Included in supplementary file.

16.How the identified compound is satisfactory compared to the one approved latestly” Siltuximab, a chimeric anti-IL-6, has recently been licensed for the 4 treatment of iMCD”

17.Author have highlighted the difference for docking with apoprotein and complex. Add a few sentences to highlight the study and any reference materials for such similar studies.

18.Author has mentioned only hydrogen bonds that are involved in the stability, but add other bonds that are involved in the interaction with the ligands.

19.The author has not mentioned the future perspective, just mentioned the ligands could be used. But inorder to differentiate and highlight their study a comparison should be made and discussed how their study is novel compared to other work

Reviewer #2: The manuscript titled “Structure-based 3D-Pharmacophore Modeling to Discover Novel Interleukin 6 Inhibitors: An In silico Screening, Molecular Dynamics Simulatons and Binding Free Energy Calculations” 3D-pharmacophore based virtual screening to identify protein-protein interaction blocking inhibitors. Overall, the work is well designed and executed, but lacking the detailed explanation in the molecular dynamics simulations. The results need to be explained in molecular detail. Most of the figures are cropped and they are not clearly presented.

6. PLOS authors have the option to publish the peer review history of their article (what does this mean?). If published, this will include your full peer review and any attached files.

Reviewer #1: **Yes: **Dr. R. Beema Shafreen

Reviewer #2: **Yes: **Konda Reddy Karnati

---

## [Author Response · Author response to Decision Letter 0]

22 Feb 2022

Dear Editor and Reviewers, 

The authors would like to thank the editor and the two reviewers for their valuable comments and questions that contribute to the quality of our work. In this document, we have revised the original manuscript following the comments suggested by the reviewers. Below are our responses to your specific comments. Changes are colored in red in our revised version.

Academic editor:

Journal Requirements: When submitting your revision, we need you to address these additional requirements. 

2. We note that you have provided funding information. However, funding information should not appear in the Acknowledgments section or other areas of your manuscript. We will only publish funding information present in the Funding Statement section of the online submission form. Please remove any funding-related text from the manuscript and let us know how you would like to update your Funding Statement. The funders had no role in study design, data collection and analysis, decision to publish, or preparation of the manuscript"

Response: Thank you for your comment. We have revised the manuscript to check for PLOS ONE's style requirements following the PLOS ONE style templates. And we have removed the funding-related text in the Acknowledgments section from the manuscript .

Reviewer #1

Authors in the present study have highlighted the importance of interleukin in several biological functions such as immune regulation, hematopoiesis, and inflammation. The pathogenic role of IL-6 during ARDS and COVID19 were highlighted. Therefore, the author has selected IL-6 as potential target and has screened for inhibitors using structure-based 3D pharmacophore modeling, MD simulation and validation through binding free energy calculation. However few questions need to be addressed and therefore I recommend a major revision of the manuscript to be suitable for publications in PLOS One.

Abstract:

1.Provide space “ membranebound” 

Response: The authors have made the correction as requested (line 6 – page 2)

2. Check spelling “inteaction” : 

Response: The authors have made the correction as requested (line 7 – page 2)

3. Revise the sentence “However, to date, no orally available small-molecule drug has been approved”. 

Response: The sentence has been revised as “To date, no orally available small-molecule drug has been approved.” (line 8 – page 2)

4. Remove “however”. 

Response: The authors have removed this word as suggested by the reviewer (line 8 – page 2)

5. Mention the key residues in () 

Response: The authors have mentioned the key residues and the sentence has been modified as “First, two pharmacophore models were constructed based on the interactions between key residues of IL-6 (Phe74, Phe78, Leu178, Arg179, Arg182) and IL-6Rα (Phe229, Tyr230, Glu277, Glu278, Phe279).” (line 10-12, page 2)

6. What is the “12 potential substances” here ligands or compounds? Please modify the sentence.

Response: The authors have modified the sentence as line 14-16,(page 2)

7. Authors should highlight the future studies or importance of the ligands identified as lead in the one sentence in the end of the abstract.

Response: The comment of the reviewer is correct. It is necessary to highlight the future studies or importance of the ligands identified as lead in the one sentence in the end of the abstract. And the last sentence of the abstract has been added as “These results will pave the way for the design and optimization of more specific compounds to combat cytokine storm in severe coronavirus patients.” (line 19-20, page 2)

Introduction:

8. Update the number of COVID cases according to the reference [7].

Response: The authors have updated the number of COVID cases according to the reference [7]. (line 7-8, page 3)

9. Modify the sentence” IL-6 activates cells by binding to IL-6 receptor (IL-6Rα) and gp130. Ternary complex forms 14 a hexamer containing IL-6, IL6Rα, and gp130, sequential and cooperative assembly”. Flaws in continuation of the sentence should be corrected.

Response: The comment of the reviewer is correct. The sentence has been modified as “IL-6 activates cells by binding to IL-6 receptor (IL-6Rα) and gp130. Ternary complex forms a hex-amer consisting of two molecules each of IL-6, IL6Rα, and gp130 chains, sequential and coopera-tive assembly.” (line 15-17, page 3)

10. What does the author mean by “biologics”. Correct it.

Response: The authors have changed the word “biologics” to “monoclonal antibodies (mAbs)” better match the content of the manuscript. (line 8, Page 4)

11. “Most previous studies mainly focused on '' please correct it. Author has to improve the grammar throughout the manuscript.

Response: The authors have modified the sentence as “Most studies mainly focused on” (Line 12-13, Page 4) and the grammar has been improved throughout the manuscript.

Materials and methods:

12. Why has the author selected “5FUC.pdb for their studies?

Response: In this study, we construct the 3D-pharmacophore models based on the protein-protein interaction (PPI pharmacophore), which were only established when a 3D structure of the PPI complex was available. Up to now, there are two X-ray protein structures containing the IL6/IL-6Rα complex in the RCSB, namely 1P9M (resolution : 3.65 Å) and 5FUC (resolution : 2.7 Å). However, at low resolution (around 3-4 Å), only the basic contours of a macromolecule backbone are observed in the density and it may be impossible to produce an atomic model with any degree of certainty [1]. Besides, the proteins with a resolution of <2.7 Å, which could see water molecules and hydrogen bond lengths clearly in the crystal structure [2], are completely suitable for selecting hydrogen bonds for the evaluation of protein-ligand interactions in this study. The manuscript has been revised as page 5 (line 11-14)

13.Why the author has studied “ adverse effects on genital organs, teratogenicity and developmental were also evaluated through the parameter Reproductive Toxicity” how this is correlated with the present study?

Response : The authors have predicted the adverse effects on genital organs, teratogenicity and developmental throughout the parameter because the reproductive toxicity is an important regulatory endpoint in health hazard assessment. The regulatory agencies around the world, including Food and Drug Administration (FDA) generally requires developmental and reproductive toxicity (DART) testing of all new drugs to be used by women of childbearing age or men of reproductive potential [3]. However, the in vivo tests are expensive, time-consuming, and require many animals, which must be killed, in silico approaches as the alternative strategies have been developed to assess the potential reproductive toxicity (reproductive toxicity) of chemicals [4]. Therefore, in this study, we conducted an assessment of the adverse effects on genital organs, teratogenicity, and developmental were also evaluated through the parameter Reproductive Toxicity by The ADMET Predictor 10.0 software or the purpose of saving time and money for future in vivo studies. The manuscript has been added as page 7 (line 9)

14.The of version of the LeadIT in line 12 can be incorporated in line 7 

Response: The authors have incorporated the version of the LeadIT in line 12 with line 7 (line 11 and line 19, respectively, page 7 at the new manuscript)

Results:

15.How the author confirmed the identified molecule is a small molecule? Just rule of five will determine the identified compound as lead? What criteria under rule of five were considered? Included in supplementary file.

Response: Small-molecule drugs are defined as chemical compounds with a molecular weight in the range of 0.1–1 kDa. They are smaller than biologics or bio-therapeutic modalities, which are generally more than 1 kDa in molecular size [5]. The small-molecule drugs are developed to follow Lipinski’s rule of five to be made bioavailable to the patient and be cleared from the body after its action. Under this law, drug-like properties must have a log P-value < 5, a molecular weight < 500 Da, H-binding acceptors (HBA) ≤ 10, and H-binding donors (HBD) ≤ 5 [ref 23]. The manuscript has been added as page 7 (line 2-4).

Besides the 5 lipinski’s rule, there are also some filters used by major pharmaceutical companies to test the drug-like properties of oral drug candidates such as Ghose (Amgen) [6], Veber (GSK) [7], Egan ( Pharmacia) [8] and Muegge (Bayer) [9]. However, in this study, we only apply the 5-lipinski rule because this filter has relatively fully met the basic criteria for filtering oral small molecule drug candidates and is also widely applied in many in silico studies. The study conducted drug-likeness with drug-like tool in MOE 2015 software and did not proceed to retrieve these results. However, in S2 table, we presented properties that met the criteria of 5 lipinski's rule of the 12 top ligands. And the manuscript has been revised as page 13 (line 3-6).

16. How the identified compound is satisfactory compared to the one approved latestly” Siltuximab, a chimeric anti-IL-6, has recently been licensed for the 4 treatment of iMCD”.

Response: According to the introduction, siltuximab is a chimeric anti-IL-6 which has recently been licensed for the treatment of iMCDO. This monoclonal antibody (mAb) also inhibits IL-6 at site I [ref 4], which coincides with the site that our study is conducting. However, mAb has several disadvantages, such as high cost, invasive use, and high immunogenicity rate. Compared to mAb–Siltuximab, the small molecular structures screened in this study have several advantages including easier oral administration, superior tissue penetration, variable pharmacokinetics, and low manufacturing costs.

17.Author have highlighted the difference for docking with apoprotein and complex. Add a few sentences to highlight the study and any reference materials for such similar studies.

Response: The authors formed the comparison of the MD trajectories between the apoprotein and the complex to simultaneously evaluate the protein fluctuations (RMSD) and the impact of ligand binding on protein residues (RMSF) in these two states. As we know, many recent studies, especially those involving MD simulation in SAR-CoV2 inhibitor studies, have applied similar comparative analysis [ref 45, 46]. The manuscript has been modified and implemented as page 14. (line 4-8).

18.Author has mentioned only hydrogen bonds that are involved in the stability, but add other bonds that are involved in the interaction with the ligands.

Response: In addition to assessing the ratio of hydrogen bonds, the study also evaluated the influence of other bonds such as salt bridge and hydrophobic interactions of the ligands on IL-6. Because, according to previous studies (Fontaine’s study and Kalai’s study), these types of binding play an important role in PPI between IL-6 and its receptor (IL-6Rα). 

19.The author has not mentioned the future perspective, just mentioned the ligands could be used. But inorder to differentiate and highlight their study a comparison should be made and discussed how their study is novel compared to other work

Response: The reviewer’s suggestion is very valuable. Up to date, there have not been any studies related to the search for small molecule structures capable of inhibiting PPI between IL-6 and IL-6Rα at site I. However, in recent studies looking for IL-6 inhibitors to acute respiratory distress syndrome in severe covid-19 patients, monoclonal antibodies such as tocilizumab, sarilumab, siltuximab emerged as a top option. These targeted monoclonal antibodies can also reduce downstream IL-6 signaling pathways by direct inhibition of the interaction between IL-6 and IL-6Rα at site I [ref 48]. Therefore, it can be seen that searching for small molecules of substances capable of inhibiting PPI between IL6/IL-6Rα at site I is a potential research direction. And the manuscript has been added as page 24 (line 11-17)

Reviewer #2

The manuscript titled “Structure-based 3D-Pharmacophore Modeling to Discover Novel Interleukin 6 Inhibitors: An in silico Screening, Molecular Dynamics Simulations and Binding Free Energy Calculations” 3D-pharmacophore based virtual screening to identify protein-protein interaction blocking inhibitors. Overall, the work is well designed and executed, but lacking the detailed explanation in the molecular dynamics simulations. The results need to be explained in molecular detail. Most of the figures are cropped and they are not clearly presented.

Response: The reviewer ’s comments are valuable. The authors added detailed analysis of the ligand-protein interactions based on the ligand molecular structure as illustrated in Fig 6 and S8 Fig. The results were discussed on pages 21-22 and put in below. The figures have been checked and presented more clearly.

Detailed analysis of the ligand-protein interactions based on the ligand molecular structure were discussed as follows: 

ZINC02997430 formed both hydrogen bonds and hydrophobic interactions with the two key residues Arg179 and Phe74 of IL-6 with high frequency. At the phenyl ring of the benzoate side branch, the ligand participated in π-alkyl and π-π interactions with Arg179 and Phe74 with a frequency of 76% and 51%, respectively. In addition, the carboxylate group (�COO-) of this branch chain also accepted hydrogen bonds from Phe74 with a frequency of 35%. At the main chain of the molecule, the nitro group (�N+OO-) formed stronger hydrogen bonds with Arg179 with a frequency 91% (Fig 6A and S8A Fig).

The structural core of molecules ZINC83804241, ZINC72026870, and ZINC46227820 contains a piperazine cyclic consisting of a six-membered ring containing two nitrogen atoms at the opposite. These N atoms (�N+) were the main agents that created the salt bridges with the two key residues Glu277 and Glu278 of IL-6Rα. In particular, the ratio of salt bridge interaction between three above ligands with Glu278 with frequency of 367% (Fig 6D and S8D Fig), 234% (Fig 6C and S8C Fig) and 145% (Fig 6B and S8B Fig), respectively. While the key residues Glu277 only formed a strong salt bridge interaction with ZINC83804241 with frequency of 148% (Fig 6D and S8D Fig). ZINC83804241 can be considered as the strongest salt-bridging ligand with IL-6Rα. Besides the salt bridge interaction, these three ligands also participate in hydrogen bonds and hydrophobic interactions with the key residues Phe229, Tyr230, Glu277, Glu278, and Phe279 of IL-6Rα. Similar to 50 ns MD simulations, the results of 100 ns MD trajectories suggest that ZINC46227820 acted as hydrogen donors to Glu278 and Phe229. The �NH groups of piperazin cyclic and the side chain donated hydrogen bonds with Phe229 and Glu278 with frequency of 193% and 77%, respectively (Fig 6B and S8B Fig). In addition, this compound also interact with Phe279 by π-π interaction at the phenyl ring of the main chain with a frequency of 51% (Fig 6B and S8B Fig). On the other hand, Glu278 and Phe229 also accepted hydrogen bonds from the �NH groups of ZINC83804241, ZINC72026870 with a high frequency of over 100%, but the phenyl groups formed a weak π-π interaction with Tyr230 and Phe279 with a very low occupancy of under 50% (Figs 6C and 6D, S8C and S8D Figs). Similar to salt bridge interaction above, Glu277 only interacted strongly with ZINC83804241 by accepting hydrogen bonds from �NH groups of piperazin ring with a frequency of 97% (Fig 6D and Fig S8D).

References

1. Zheng H, Hou J, Zimmerman MD, Wlodawer A, Minor W. The future of crystallography in drug discovery. Expert Opinion on Drug Discovery. 2014;9(2):125-37. doi: 10.1517/17460441.2014.872623.

2. Wlodawer A, Minor W, Dauter Z, Jaskolski M. Protein crystallography for non-crystallographers, or how to get the best (but not more) from published macromolecular structures. The FEBS journal. 2008;275(1):1-21. doi: 10.1111/j.1742-4658.2007.06178.x. PubMed Central PMCID: PMC4465431.

3. Faqi ASJSBiRM. A critical evaluation of developmental and reproductive toxicology in nonhuman primates. 2012;58(1):23-32.

4. Jiang C, Yang H, Di P, Li W, Tang Y, Liu GJJoAT. In silico prediction of chemical reproductive toxicity using machine learning. 2019;39(6):844-54.

5. Chhabra M. Chapter 6 - Biological therapeutic modalities. In: Hasija Y, editor. Translational Biotechnology: Academic Press; 2021. p. 137-64.

6. Ghose AK, Viswanadhan VN, Wendoloski JJ. A knowledge-based approach in designing combinatorial or medicinal chemistry libraries for drug discovery. 1. A qualitative and quantitative characterization of known drug databases. Journal of combinatorial chemistry. 1999;1(1):55-68. Epub 2000/04/04. doi: 10.1021/cc9800071. PubMed PMID: 10746014.

7. Veber DF, Johnson SR, Cheng HY, Smith BR, Ward KW, Kopple KD. Molecular properties that influence the oral bioavailability of drug candidates. Journal of medicinal chemistry. 2002;45(12):2615-23. Epub 2002/05/31. doi: 10.1021/jm020017n. PubMed PMID: 12036371.

8. Egan WJ, Merz KM, Jr., Baldwin JJ. Prediction of drug absorption using multivariate statistics. Journal of medicinal chemistry. 2000;43(21):3867-77. doi: 10.1021/jm000292e. PubMed PMID: 11052792.

9. Muegge I, Heald SL, Brittelli D. Simple selection criteria for drug-like chemical matter. Journal of medicinal chemistry. 2001;44(12):1841-6. doi: 10.1021/jm015507e. PubMed PMID: 11384230.

---

## [Decision Letter · Decision Letter 1]

24 Mar 2022

Structure-Based 3D-Pharmacophore Modeling to Discover Novel Interleukin 6 Inhibitors: An In silico Screening, Molecular Dynamics Simulations and Binding Free Energy Calculations.

PONE-D-21-38021R1

Dear Dr. Thai,

We’re pleased to inform you that your manuscript has been judged scientifically suitable for publication and will be formally accepted for publication once it meets all outstanding technical requirements.

Kind regards,

Chandrabose Selvaraj, Ph.D.

Academic Editor

PLOS ONE

Additional Editor Comments (optional):

Reviewers' comments:

Reviewer's Responses to Questions

6. Review Comments to the Author

Reviewer #2: (No Response)

Reviewer #3: All the corrections were carried out meticulously hence the manuscript can be accepted in its current form for publication

---

## [Editor Report · Acceptance letter]

28 Mar 2022

PONE-D-21-38021R1 

Structure-Based 3D-Pharmacophore Modeling to Discover Novel Interleukin 6 Inhibitors: An In silico Screening, Molecular Dynamics Simulations and Binding Free Energy Calculations 

Dear Dr. Thai:

I'm pleased to inform you that your manuscript has been deemed suitable for publication in PLOS ONE. Congratulations! Your manuscript is now with our production department. 

Kind regards, 

on behalf of

Dr. Chandrabose Selvaraj 

Academic Editor

PLOS ONE